# Adenosine A_2A_ and A_3_ Receptors Are Able to Interact with Each Other. A Further Piece in the Puzzle of Adenosine Receptor-Mediated Signaling

**DOI:** 10.3390/ijms21145070

**Published:** 2020-07-17

**Authors:** Alejandro Lillo, Eva Martínez-Pinilla, Irene Reyes-Resina, Gemma Navarro, Rafael Franco

**Affiliations:** 1Department of Biochemistry and Physiology, Faculty of Pharmacy and Food Science, University of Barcelona, 08028 Barcelona, Spain; alilloma55@gmail.com (A.L.); ire-reyes@hotmail.com (I.R.-R.); g.navarro@ub.edu (G.N.); 2Department of Morphology and Cell Biology, Faculty of Medicine, University of Oviedo, 33006 Asturias, Spain; martinezpinillaeva@gmail.com; 3Instituto de Neurociencias del Principado de Asturias (INEUROPA), 33006 Asturias, Spain; 4Instituto de Investigación Sanitaria del Principado de Asturias (ISPA), 33006 Asturias, Spain; 5Centro de Investigación Biomédica en Red Enfermedades Neurodegenerativas (CIBERNED), 28031 Madrid, Spain; 6Molecular Neurobiology Laboratory, Department of Biochemistry and Molecular Biomedicine, University of Barcelona, 08028 Barcelona, Spain; 7School of Chemistry, Universitat de Barcelona, Diagonal 645, 08028 Barcelona, Spain

**Keywords:** cross-antagonism, cortical neurons, heteromer print, purinergic P1 receptors, G-protein-coupled receptors

## Abstract

The aim of this paper was to check the possible interaction of two of the four purinergic P1 receptors, the A_2A_ and the A_3_. Discovery of the A_2A_–A_3_ receptor complex was achieved by means of immunocytochemistry and of bioluminescence resonance energy transfer. The functional properties and heteromer print identification were addressed by combining binding and signaling assays. The physiological role of the novel heteromer is to provide a differential signaling depending on the pre-coupling to signal transduction components and/or on the concentration of the endogenous agonist. The main feature was that the heteromeric context led to a marked decrease of the signaling originating at A_3_ receptors. Interestingly from a therapeutic point of view, A_2A_ receptor antagonists overrode the blockade, thus allowing A_3_ receptor-mediated signaling. The A_2A_–A_3_ receptor heteromer print was detected in primary cortical neurons. These and previous results suggest that all four adenosine receptors may interact with each other. Therefore, each adenosine receptor could form heteromers with distinct properties, expanding the signaling outputs derived from the binding of adenosine to its cognate receptors.

## 1. Introduction

The laboratories of Susan George and Lakshmi Devi were the first to demonstrate that two G-protein-coupled receptors (GPCRs), for the same neurotransmitter, may form functional heterodimers. In fact, receptor heteromers composed of different opioid receptor types were reported circa 20 years ago [1,2,3,4]. Similarly, dopamine D_1_–D_2_ receptor heteromers were described with a particular feature, namely coupling to G_q_. Whereas D_1_ couples to G_s_ and D_2_ couples to G_i_, the D_1_–D_2_ heterodimer couples to G_q_, that is, neurotransmission mediated by this specific heteromer does not course with changes of intracellular cAMP levels, but of cytoplasmic Ca^2+^ concentrations [5,6,7,8]. Therefore, receptor heteromers are considered novel functional units acting differently than individually expressed receptors [9,10,11].

In the adenosine receptor field, the first reported heteromer resulted from a positive result in a supposedly negative control [12]. There are four types of adenosine receptors, all belonging to the GPCRs family. The A_1_ receptor (A_1_R) and A_3_ receptor (A_3_R) couple to G_i_, whereas the A_2A_ receptor (A_2A_R) and the A_2B_ receptor (A_2B_R) couple to G_s_. Moreover, A_1_R and A_2A_R show high affinity for adenosine, and the A_2B_R has the lowest affinity for the nucleoside. Although it was known that adenosine may regulate calcium ion concentration in the cytoplasm of, among others, lymphocytes, myocytes, and chondrogenic progenitor cells [13,14,15], to our knowledge, no report had identified G_q_ proteins as coupled to adenosine receptors. Therefore, it was not expected that the A_1_R could interact with the A_2A_R and couple to G_q_ as in the D_1_–D_2_ receptor heteromer. Unexpectedly, they interact and the physiological role is quite relevant in different cell types, neurons and astrocytes included. The heteromer is composed of two molecules of the A_1_R, two molecules of the A_2A_R, one G_s_ protein, and one G_i_ protein. Thus, the functional unit, whose 3D structure has been proposed for the first time for a GPCR heteromer complex, acts via A_1_R–G_i_ at low adenosine concentrations and via A_2A_R–G_s_ when adenosine level increases. The long C-terminal domain of the A_2A_R intermingles within the A_1_R and blocks G_i_ engagement upon receptor activation [16,17]. As a consequence, A_1_R–A_2A_R heteromer regulates neurotransmitter release in glutamatergic terminals and the uptake of gamma-amino butyric acid in astrocytes; this regulation is totally opposite at low versus at high adenosine concentrations [12,18]. Overall, the heteromer is an adenosine concentration sensing device whose dual role could not be achieved by the individual operation of the two receptors.

Another example of interaction among adenosine receptor types is the A_2A_R–A_2B_R heteromer. On the one hand, the expression of the A_2B_R, that is, the amount of heteromers, regulates A_2A_R-mediated signaling. The higher the number of heteromers, the lesser the A_2A_R-mediated signaling, even keeping constant the quantity of A_2A_R on the cell surface [19,20]. Notice that this property of the heteromer is counterintuitive as both protomers in the heteromer are coupled to G_s_ and, therefore, their activation was supposed to result in potentiation of G_s_ engagement. Also interesting is the recently reported role of the A_2A_R–A_2B_R heteromer in the control of energy expenditure in adipose tissue (see details in [21]).

In recent years, the existence of the heteromer formed by the A_1_R and the A_3_R was suggested on the basis of allosteric modulations [22]. Apart from confirming the direct interaction of these two receptors, this observation underscores that few adenosine receptor–receptor heteromers remain to be discovered. The aim of this study was to test the potential interaction of the G_i_-coupled A_3_R and the G_s_-coupled A_2A_R. The heteromer was identified in a heterologous expression system and in primary cultures of mouse neurons. Finally, properties of the complex in terms of signaling were assessed.

## 2. Results

### 2.1. Molecular Interaction between A_2A_R and A_3_R

Adenosine receptors are divided into high affinity A_1_R and A_2A_R types, and low affinity A_2B_R and A_3_R types [23,24]. Members of this family interact to form heteromeric complexes, however, this possibility has not yet explored for the A_2A_R and A_3_R types. To first address a possible interaction between these two receptors, we performed immunocytochemical assays to look for colocalization. We used a heterologous system consisting of human embryonic kidney HEK-293T cells, which were transfected with A_3_R fused to yellow fluorescent protein (YFP), A_2A_R fused to Rluc, or both. The results in Figure 1A show a marked colocalization of the signal of the two receptors; the label was mainly in the cell surface (notice that the planes of the confocal images are taken near the surface of the slide) and did not substantially change by previous treatment with agonists. Because of the fact that this technique cannot demonstrate a direct interaction, we transfected HEK-293T cells with a constant amount of the cDNA for A_2A_R–Rluc and increasing quantities of the cDNA for A_3_R–YFP, then bioluminescence resonance energy transfer (BRET) was determined. Interestingly, a saturation BRET curve indicating a specific interaction between A_2A_R and A_3_R was obtained, as shown in Figure 1B. The calculated parameters were BRET_max_ = 51 ± 4 mBU and BRET_50_ = 12 ± 3. As negative control, the same assay was developed with a constant amount of A_2A_R–Rluc and increasing quantities of D_4_R–YFP, obtaining a nonspecific linear signal that indicates a lack of interaction between these receptors (Figure 1B). In summary, co-expression of the two receptors in a heterologous system leads to A_2A_R–A_3_R heteromer (A_2A_R/A_3_R-Het) formation. Taking into account previous knowledge, receptor heteromers are formed in the endoplasmic reticulum and traffic to the cell surface, where they complex with heteromeric G proteins to acquire further stabilization. Indeed, the final structure is significantly modified by agonist binding [16,17,19,21].

### 2.2. Effect of the A_3_R Expression on Homogeneous Ligand Binding to the A_2A_R

A_2A_R and A_3_R are GPCRs that, upon activation by the endogenous agonist, adenosine, initiate diverse signaling pathways. Thus, the analysis of ligand binding to receptors in a heteromeric context becomes greatly relevant. Accordingly, competition experiments were performed using a homogenous assay as described in Methods. HEK-293T cells expressing HALO–A_2A_R labeled with Lumi4-Tb were incubated with 20 nM of a fluorophore-conjugated selective A_2A_R antagonist (red SCH 442416) and increasing concentrations (0–10 µM) of the A_2A_R agonist, CGS 21680. As observed in Figure 2A, CGS 21680 competed the binding of red SCH 442416 in a monophasic fashion and with a K_i_ value in the nanomolar range (35 ± 1 nM), which matches with values obtained using radiolabeled ligands. At present, there is no selective fluorophore-conjugated agonist to perform homogenous assays of binding to HALO–A_3_R expressing cells. Therefore, we measured the competition of the binding of red SCH 442416 to cells co-expressing HALO–A_2A_R and A_3_R (Figure 2B). The K_i_ value for CGS 21680, used as competitor, was one order of magnitude higher (427 ± 4 nM), thus suggesting that, when A_2A_R forms complexes with A_3_R, the structure of the orthosteric site of the A_2A_R is modified; it cannot be ruled out that this apparent increase in affinity is in part owing to CGS 21680 binding to the A_3_R.

### 2.3. Functional Characterization of A_2A_R/A_3_R-Hets in HEK-293T Cells

After detecting the existence of A_2A_R/A_3_R-Hets in co-transfected HEK-293T cells, we questioned the functional implication of this newly discovered protein–protein interaction. Thanks to A_2A_R coupling to G_s_ protein, agonists such as CGS 21680 lead to adenylate cyclase activation and to increased cytosolic cAMP levels. In contrast, A_3_R couples to G_i_ and its activation leads to decreased cytosolic cAMP concentrations. Taking into account these facts, confirmed in cells transfected with the plasmid for one of the two receptors, we measured cAMP levels in HEK-293T cells transfected with cDNAs for A_2A_R (0.3 µg) and for A_3_R (0.4 µg), and treated with selective agonists. While CGS 21680 stimulation induced a marked rise in cAMP concentration, IB-MECA had no significant effect over forskolin-induced increase in cAMP levels (Figure 3A). This result indicates that A_3_R–G_i_ coupling is blocked in the heteromeric context, which can be considered a print to detect A_2A_R/A_3_R-Hets in native tissues/cells. When cells were simultaneously treated with the two agonists, the effect was similar to that obtained upon A_2A_R activation, thus reinforcing the hypothesis that A_2A_R stimulation blocks A_3_R induced signaling. In the case of cells pretreated with the selective antagonists (SCH 442416 for the A_2A_R or PSB-10 for the A_3_R), we found that SCH 442416 blocked CGS 21680-induced effects, while it potentiated the G_i_-mediated effect elicited by IB-MECA. These results suggest that antagonist binding to A_2A_R leads to a structural reorganization in the A_2A_R/A_3_R-Hets that blunts the A_2A_R-mediated blockade of A_3_R–G_i_ coupling (Figure 3A). For its part, the A_3_R antagonist (PSB-10) had no effect on A_2A_R activation; no cross-antagonism of A_3_R over A_2A_R was detected.

Next, β-arrestin 2 recruitment was analyzed in cells expressing A_2A_R/A_3_R-Hets and the findings were similar to those obtained in cAMP determination assays obtained in the absence of forskolin (Figure 3C). In contrast, the results of ERK1/2 phosphorylation showed significant responses produced by either the A_2A_R or the A_3_R agonist (Figure 3B). Interestingly, when the receptors were simultaneously exposed to the two agonists, ERK1/2 phosphorylation was milder than in individual receptor engagement. This phenomenon may be considered as negative crosstalk. On the one hand, when cells were pretreated with the selective antagonist for the A_2A_R (SCH 442416), a complete blockade of CGS 21680-induced MAPK activation was observed, while the antagonist was ineffective on A_3_R activation. On the other hand, pretreatment with the A_3_R antagonist (PSB-10) induced a partial cross-antagonism on A_2A_R-induced ERK1/2 phosphorylation. Dynamic mass redistribution (DMR) is a label-free technique widely used in drug discovery, especially in the field of GPCRs, that serves to analyze cell responses in the absence of any exogenous reagent (apart from receptor ligands). The DMR equipment detects changes upon time of the wavelength of light reflected by cells; picometer shifts in the wavelength of photons occur when a GPCR is activated on the cell surface [25]. DMR responses showed that the signal due to A_3_R activation was blocked when the A_2A_R was co-expressed. Furthermore, a cross-antagonism was detected, that is, the A_2A_R-induced signal was blocked by pretreatment with either A_2A_R or A_3_R antagonists (Figure 3D).

### 2.4. Discovery of A_2A_R/A_3_R-Hets in Primary Cultures of Cortical Neurons 

We moved to a more physiological environment to check whether the A_2A_R/A_3_R-Hets may be expressed in a natural source. It is known that the two adenosine receptors are expressed in different areas of the central nervous system. Owing to the implication of the adenosine receptor in neuromodulation, we addressed the possible expression of A_2A_R/A_3_R-Hets in cortical neurons by detecting the heteromer print.

Primary cultures of cortical neurons were prepared and cAMP determination and ERK1/2 phosphorylation assays were performed. The results in Figure 4A show that the selective A_3_R antagonist (PSB-10) did not counteract the effect of the selective A_2A_R agonist (CGS 21680); it is one of the features detected in HEK-293T cells. The release of the brake on A_3_R-mediated signaling by selective A_2A_R antagonists (Figure 4B) and the cross antagonism in the link to the MAPK signaling pathway (Figure 4C) were also detected. In summary, these data constitute strong evidence of A_2A_R/A_3_R-Hets expression in primary cultures of cortical neurons.

## 3. Discussion

This paper discovers a new complex formed by two different adenosine receptors that may be expressed in a heterologous system, but also in primary cultures of cortical neurons. The heteromer print is quite unique as antagonists of the A_2A_R enhance A_3_R-mediated signaling.

Especially unexpected was the discovery of heteromers formed by two receptors of the same neurotransmitter/hormone that are in complex with different cognate (heterotrimeric) G proteins. One possibility is a shift in the G-protein-coupling. In fact, the dopamine D_1_–D_2_ heteroreceptor does not couple to G_s_ nor to G_i_, but to G_q_ [5,6,7,8]. However, in the case of the A_1_R–A_2A_R heteromer, there is no shift in G-protein coupling, but the overall structure allows increasing or decreasing cAMP levels depending on the concentration of the endogenous agonist. Adenosine preferentially activates the A_1_R, thus engaging G_i_ proteins. Nevertheless, when adenosine level increases and the A_2A_R is activated within the heteromer, G_i_-coupling is blunted and G_s_ engagement occurs. The molecular determinants that make this possible are detailed elsewhere [16,17], but it is worth mentioning that the C-terminal domain of the A_2A_R plays a fundamental role.

The A_2B_R is the most enigmatic adenosine receptor; it has a very reduced affinity for the nucleoside and it is also the receptor for netrin, which belongs to a family of proteins involved in axon guidance [26]. On the one hand, comparison of the structural arrangement of A_2A_R and A_2B_R has led to the discovery that the second extracellular loop determines low (A_2B_R) or high (A_2A_R) affinity for adenosine [27]. On the other hand, as earlier commented, the expression of the A_2B_R reduces the A_2A_R-mediated functions by means of the formation of A_2A_–A_2B_ heteroreceptor complexes. Thus, it seems that the A_2B_R, *per se* or negatively acting on the A_2A_R, contributes to reducing the actions derived from extracellular adenosine accumulation.

The properties of the heteromer described here are different from those previously defined for adenosine receptor heterocomplexes. A simple scheme summarizing the operation of A_2A_R/A_3_R-Hets is provided in Figure 5. One of the features of A_2A_R/A_3_R-Hets is common to several heteromers, namely cross-antagonism [9,28,29]. However, the most important characteristic is that the A_3_R functionality is negligible within the A_2A_R/A_3_R-Het. Importantly, selective A_2A_R antagonists release the brake on A_3_R activation. This finding adds a new piece in the puzzle of both purinergic and GPCR-heteromer-mediated signaling. This specific feature may be of interest in drug discovery in a time when adenosine receptors are gaining momentum after the approval (in Japan and the USA) of istradefylline, a selective A_2A_R antagonist, as adjuvant therapy in Parkinson’s disease [30,31,32,33,34].

## 4. Materials and Methods

### 4.1. Reagents

CGS 21680 hydrochloride, IB-MECA, SCH 442416 and PSB-10 hydrochloride were purchased from Tocris Bioscience (Bristol, UK). Forskolin was purchased from Sigma-Aldrich (St. Louis, MO, USA). For non-radioactive binding assays, tag-lite buffer (TLB) was obtained from Cisbio Bioassays (LABMED, Codolet, France). The Tb derivative of O6-benzylguanine is commercialized as HaloTag-Lumi4-Tb (SHALOTBC, Cisbio Bioassays, LABMED, Codolet, France). A_2A_R antagonist (SCH 442416 derivate) conjugated to a red fluorescent probe was purchased from Cisbio Bioassays (L0058RED, LABMED, Codolet, France). The intellectual property of TLB components and of fluorophore structure is owned by Cysbio Bioassays.

### 4.2. Neuronal Primary Cultures

By the current legislation, obtaining protocol approval is not needed if animals are sacrificed to obtain a specific tissue. CD-1 strain mice handling, sacrifice, and further experiments were conducted according to the guidelines set in Directive 2010/63/EU of the European Parliament and the Council of the European Union that is enforced in Spain by National and Regional organisms; the 3R rule (replace, refine, reduce) for animal experimentation was also taken into account. Primary cultures of cortical neurons were obtained from 19-day embryos. Cells were isolated as described in [35] and plated at a confluence of 40,000 cells/0.32 cm^2^. Cells were maintained for 14 days in Neurobasal medium supplemented with 2 mM L-glutamine, 100 U/mL penicillin/streptomycin, and 2% (*v*/*v*) B27 supplement (Gibco, Paisley, Scotland, UK), in six-well plates for functional assays.

### 4.3. Cell Culture and Transient Transfection

HEK-293T cells were grown in DMEM medium (Gibco, Paisley, Scotland, UK) supplemented with 2 mM L-glutamine, 100 U/mL penicillin/streptomycin, MEM non-essential amino acids solution (1/100), and 5% (*v*/*v*) heat inactivated fetal bovine serum (FBS) (Invitrogen, Paisley, Scotland, UK). Cells were maintained in a humid atmosphere of 5% CO_2_ at 37 °C, and were transiently transfected with plasmids for fusion proteins containing sequences encoding for human A_2A_R (Gene ID: 135; GenBank: X68486.1) and/or human A_3_R (Gene ID: 140; GenBank: B029831.1) using the PEI (polyethylenimine, Sigma-Aldrich, St. Louis, MO, USA) method, as previously described [17,36]. Cells were used 48 h later for functional assays.

### 4.4. Fusion Proteins and Expression Vectors

The human cDNAs for the A_2A_R and A_3_R cloned in pcDNA3.1 were amplified without their stop codons using sense and antisense primers harboring either unique BamHI and XhoI sites for A_3_R and Hind III and BamHI sites for A_2A_R. The fragments were subcloned to be in frame with the sequence coding for an enhanced YFP (pEYFP-N1; Clontech, Heidelberg, Germany) or a Rluc protein (pRluc-N1, PerkinElmer, Wellesley, MA, USA). Final cDNAs encoding for A_2A_R–YFP, A_2A_R–Rluc and A_3_R–YFP, and fusion proteins having the receptor at the N-terminal end.

After transfection with the corresponding plasmids, the health and viability of transfected cells were proved using the appropriate negative controls. In addition, expression of receptors was tested by either fluorescent confocal microscopy (Figure 1A) or Rluc expression, and receptor function was tested by performing ERK1/2 activation assays. Using Trypan Blue solution (T8154, Sigma-Aldrich, St. Louis, MO, USA), the percentage of non-viable cells when collected for experiment performance was <15.

### 4.5. Bioluminescence Resonance Energy Transfer (BRET) Assays

HEK-293T cells growing in six-well plates were transiently co-transfected with a constant amount of cDNA encoding for A_2A_R–Rluc and with increasing concentrations of cDNAs for A_3_R–YFP or D_4_R–YFP as negative control. Forty-eight hours post-transfection, cells were washed twice in Hank’s Balanced Salt Solution (HBSS; 137 mM NaCl, 5 mM KCl, 0.34 mM Na_2_HPO_4_, 0.44 mM KH_2_PO_4_, 1.26 mM CaCl_2_, 0.4 mM MgSO_4_, 0.5 mM MgCl_2_, and 10 mM HEPES, pH 7.4) supplemented with 0.1% glucose (*w*/*v*), detached by gently pipetting and resuspended in the same buffer. To assess the number of cells per plate, protein concentration was determined using a Bradford assay kit (Bio-Rad, Munich, Germany) with bovine serum albumin dilutions as standards. To quantify YFP-fluorescence expression, cell suspension (20 μg protein) was distributed in 96-well microplates (black plates with a transparent bottom, Porvair, Leatherhead, UK). Fluorescence was read using a Mithras LB 940 (Berthold, Bad Wildbad, Germany) equipped with a high-energy xenon flash lamp, using a 10 nm bandwidth excitation and emission filters at 485 nm and 530 nm, respectively. YFP-fluorescence expression was determined as the fluorescence of the sample minus the fluorescence of cells only expressing protein-Rluc. For BRET measurements, the equivalent to 20 μg cell suspension was distributed in 96-well microplates (white plates, Porvair, Leatherhead, UK), and 5 μM coelenterazine H was added (PJK GMBH, Kleinblittersdorf, Germany). One minute after coelenterazine H addition, the readings were collected using a Mithras LB 940 (Berthold, Bad Wildbad, Germany), which allowed the integration of the signals detected in the short-wavelength filter at 485 nm (440–500 nm) and the long-wavelength filter at 530 nm (510–590 nm). To quantify receptor-Rluc expression, luminescence readings were collected 10 min after addition of 5 μM coelenterazine H. The net BRET is defined as [(long-wavelength emission)/(short-wavelength emission)]-Cf, where Cf corresponds to [(long-wavelength emission)/(short-wavelength emission)] for the Rluc construct expressed alone in the same experiment. Data in BRET curves that depict an equilateral hyperbola were fitted by a non-linear regression equation using the GraphPad Prism software (San Diego, CA, USA). BRET values for specific interactions are given as milli BRET units (mBU: 1000 × net BRET).

### 4.6. Immunocytochemistry

Transfected HEK-293T cells expressing A_3_R–YFP and/or A_2A_R–Rluc were treated with vehicle, with 100 nM CGS 21680, or with 100 nM IB-MECA for 30 min. Then, cells were fixed in 4% paraformaldehyde for 15 min and washed twice with PBS containing 20 mM glycine before permeabilization with the same buffer containing 0.2% Triton X-100 (5 min incubation). Cells were treated for 1 h with PBS containing 1% bovine serum albumin, labeled with a mouse anti-Rluc (1/100, MAB4400, Millipore, Merck, Darmstadt, Germany) antibody, and subsequently treated with a Cy3-conjugated anti-mouse (1/200, 715-166-150 (red), Jackson ImmunoResearch, St. Thomas Place, UK) IgG secondary antibody (1 h each). Samples were washed several times and mounted with 30% Mowiol (Calbiochem, Merck, Darmstadt, Germany). Samples were observed under a Leica SP2 confocal microscope (Leica Microsystems, Wetzlar, Germany).

### 4.7. Homogeneous Competition Binding Assays in Living Cells

HEK-293T cells expressing HALO-tagged A_2A_R in the presence or in the absence of A_3_R were seeded in six-well plates. After 48 h, medium was removed and cells were treated with 100 nM HALOTag-Lumi4-Tb, previously diluted in 3 mL TLB 1X, for 1 h at 37 °C under 5% CO_2_ atmosphere in a cell incubator. Cells were then washed four times with 2 mL TLB 1X to remove the excess of HALOTag-Lumi4-Tb, detached with enzyme-free cell dissociation buffer, centrifuged 5 min at 1500 rpm, and collected in 1 mL TLB 1X. Cells at densities in the 2500–3000 cells/well range were plated in white opaque 384-well plates (10 µL). Then, 5 μL of 20 nM fluorophore-conjugated A_2A_R antagonist were added in the presence of vehicle or of CGS 21680 hydrochloride at increasing concentrations (0–10 μM range, 5 μL total volume). Plates were then placed at room temperature for 2 h before signal detection. Detailed description of the HTRF assay is found in [37]. Signal was detected using an PheraSTAR microplate reader (PerkinElmer, Waltham, MA, USA) equipped with a Fluorescence Resonance Energy Transfer (FRET) optic module allowing donor excitation at 337 nm and signal collection at both 665 and 620 nm. A frequency of 10 flashes/well was selected for the xenon flash lamp excitation. The signal was collected at both 665 and 620 nm using the following time-resolved settings: delay: 150 μs and integration time: 500 μs. HTRF ratios were obtained by dividing the acceptor (665 nm) by the donor (620 nm) signals and multiplying by 10,000. The 10,000-multiplying factor is used solely for the purpose of easier data handling.

### 4.8. cAMP Determination

For cAMP studies, HEK-293T transfected cells and cortical neurons were prepared. Signaling experiments were performed as previously described [16,17,19,38]. Briefly, 2 h before initiating the experiment, cell-culture medium was replaced by serum-free DMEM medium. Then, cells were detached, resuspended in growing medium containing 50 μM zardaverine, and placed in 384-well microplates (2500 cells/well). Cells were pretreated (15 min) with antagonists (SCH 442416 for A_2A_R and/or PSB-10 for A_3_R) and stimulated with agonists (CGS 21680 for A_2A_R and/or IB-MECA for A_3_R) (15 min) before adding 0.5 μM forskolin or vehicle. Readings were performed after 1 h incubation at 25 °C. HTRF energy transfer measures were performed using the Lance Ultra cAMP kit (PerkinElmer, Waltham, MA, USA). Fluorescence at 665 nm was analyzed in a PHERAstar Flagship microplate reader equipped with an HTRF optical module (BMG Lab Technologies, Offenburg, Germany).

### 4.9. ERK Phosphorylation Assays

To determine ERK1/2 phosphorylation, HEK-293T transfected cells and cortical neurons were seeded at a density of 40,000 cells/well in transparent Deltalab 96-well microplates, and kept at the cell incubator. Two hours prior to the experiment, the medium was substituted by serum-free DMEM medium. Then, cells were pre-treated for 10 min at 25 °C with antagonists (SCH 442416 for A_2A_R and/or PSB-10 for A_3_R) in serum-free DMEM medium and stimulated for an additional 7 min with the selective agonists (CGS 21680 for A_2A_R and/or IB-MECA for A_3_R). Then, cells were washed twice with cold PBS before the addition of lysis buffer (15 min treatment). Then, 10 μL of each supernatant was placed in white ProxiPlate 384-well microplates and ERK1/2 phosphorylation was determined using AlphaScreen^®^SureFire^®^ kit (Perkin Elmer, Waltham, MA, USA), following the instructions of the supplier and using an EnSpire^®^ Multimode Plate Reader (PerkinElmer, Waltham, MA, USA).

### 4.10. Dynamic Mass Redistribution (DMR) Assays

Cell mass redistribution induced upon receptor activation was detected by illuminating the underside of a biosensor with polychromatic light and measuring the changes in the wavelength of the reflected monochromatic light. The magnitude of wavelength shifts (in picometers) is directly proportional to the amount of DMR. Transfected HEK-293T cells were seeded in 384-well sensor microplates to obtain 70–80% confluent monolayers constituted by approximately 10,000 cells per well. Prior to the assay, cells were washed twice with assay-buffer (HBSS with 20 mM HEPES, pH 7.15) and incubated for 2 h with assay-buffer containing 0.1% dimethyl sulfoxide (DMSO) (24 °C, 30 μL/well). Hereafter, the sensor plate was scanned and a baseline optical signature was recorded for 10 min before adding 10 μL of antagonist solution (SCH 442416 for A_2A_R and/or PSB-10 for A_3_R) for 30 min, followed by the addition of 10 μL of agonist solution (CGS 21680 for A_2A_R and/or IB-MECA for A_3_R). The label-free signature was determined using an EnSpire^®^ Multimode Plate Reader (PerkinElmer, Waltham, MA, USA). DMR responses were monitored for at least 5000 s. The results were analyzed using EnSpire^®^ Workstation Software v 4.10 (PerkinElmer, Waltham, MA, USA).

### 4.11. β-arrestin 2 Recruitment

β-arrestin 2 recruitment was determined as previously described [17,36]. Briefly, BRET experiments were performed in HEK-293T cells 48 h after transfection with the cDNA corresponding to A_2A_R–YFP or A_3_R–YFP (0.5 μg cDNA each) and 1 μg cDNA corresponding to β-arrestin 2-Rluc. Cells (20 μg protein) were distributed in 96-well microplates (Corning 3600, white plates with white bottom) and were incubated with antagonists (SCH 442416 for A_2A_R and/or PSB-10 for A_3_R) for 15 min and stimulated with agonists (CGS 21680 for A_2A_R and/or IB-MECA for A_3_R) for 10 min prior to the addition of 5 μM coelenterazine H (Molecular Probes, Eugene, OR, USA). One minute after coelenterazine H addition, BRET between β-arrestin 2-Rluc and receptor-YFP was determined and quantified. The readings were collected using a Mithras LB 940 (Berthold Technologies, Bad Wildbad, Germany) that allows the integration of the signals detected in the short-wavelength filter at 485 nm and the long-wavelength filter at 530 nm. To quantify protein-Rluc expression, luminescence readings were also performed 10 min after adding 5 μM coelenterazine H.

### 4.12. Data Handling and Statistical Analysis

Data from homogeneous binding assays were analyzed using Prism 6 (GraphPad Software, Inc., San Diego, CA, USA). K_i_ values were determined according to the Cheng and Prusoff equation with K_D_ = 20 nM for A_2A_R red antagonist [39]. Signal-to-background (S/B ratio) calculations were performed by dividing the mean of the maximum value (μ_max_) by that of the minimum value (μ_min_) obtained from the sigmoid fits.

The data are mainly shown as the mean ± S.E.M. Statistical analysis was performed with SPSS 18.0 software. The test of Kolmogorov–Smirnov with the correction of Lilliefors was used to evaluate normal distribution and the test of Levene was used to evaluate the homogeneity of variance. Significance was analyzed by one-way ANOVA, followed by Bonferroni’s multiple comparison *post hoc* test. Significant differences were considered when *p* < 0.05.

## Figures and Tables

**Figure 1 ijms-21-05070-f001:**
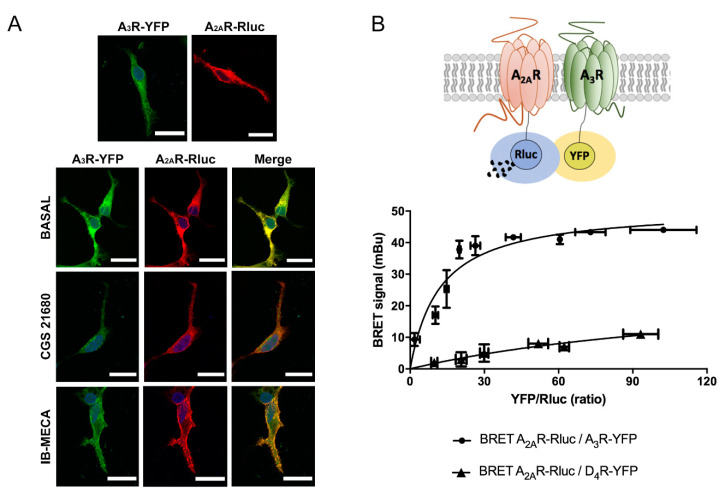
(**A**) Immunocytochemical assays were performed in HEK-293T cells expressing A_2A_R–Rluc (0.5 µg cDNA), which was detected by a mouse anti-Rluc antibody and a secondary anti-mouse Cy3 conjugated antibody (red), and/or A_3_R–YFP (0.5 µg cDNA), which was detected by its own fluorescence (green). Colocalization is shown in yellow. Cells were previously treated with 100 nM CGS 21680, 100 nM IB-MECA, or vehicle. Cell nuclei were stained with Hoechst (blue). Scale bar: 20 µm. (**B**) A_2A_R and A_3_R interact in transfected HEK-293T cells. BRET assays were performed in HEK-293T cells transfected with a constant amount of cDNA for A_2A_R–Rluc (0.1 µg) and increasing concentrations of cDNA for A_3_R–YFP (0.1 to 1 µg) or D_4_R–YFP (0.1 to 1 µg) for negative control. Values are the mean ± S.E.M. (n = 6 in triplicates). mBU: milliBRET units.

**Figure 2 ijms-21-05070-f002:**
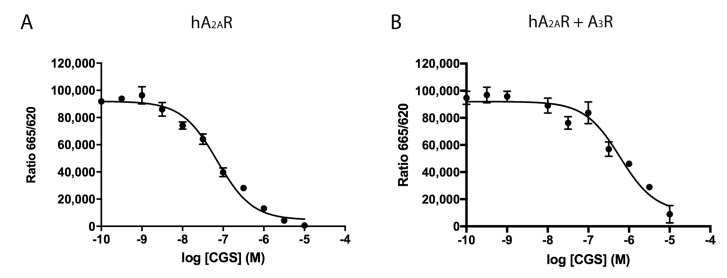
Competition experiments of SCH 442416 red ligand binding to living HEK-293T cells expressing A_2A_R or A_2A_R/A_3_R-Hets. HEK-293T cells were transfected with 1 µg cDNA for HALO–A_2A_R in the presence (**B**) or in the absence (**A**) of 0.5 µg cDNA for A_3_R. Competition binding curves were obtained by Homogeneous Time Resolved Fluorescence (HTRF) using 20 nM red SCH 442416 with increasing concentrations of CGS 21680 (A_2A_R agonist) (0–10 µM). Data are the mean ± S.E.M. of a representative experiment performed in triplicates. HTRF ratio = 665 nm acceptor signal/620 nm donor signal × 10,000.

**Figure 3 ijms-21-05070-f003:**
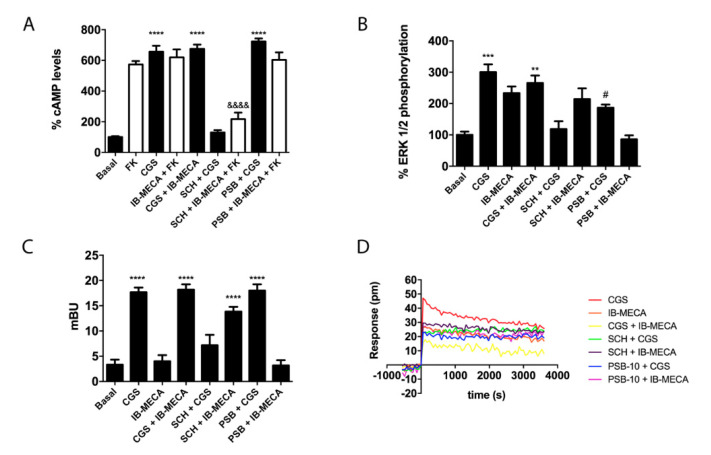
Signaling assays in HEK-293T cells expressing A_2A_R/A_3_R-Hets. HEK-293T cells expressing A_2A_R (0.3 µg cDNA) and A_3_R (0.4 µg cDNA) were pretreated with selective receptor antagonists (1 µM SCH 442416 for A_2A_R or 1 µM PSB-10 for A_3_R) and subsequently treated with selective agonists (100 nM CGS 21680 for A_2A_R or 100 nM IB-MECA for A_3_R). cAMP accumulation (**A**) was determined by HTRF as described in Methods. When indicated, cells were subsequently treated for 15 min with 0.5 µM forskolin. ERK1/2 phosphorylation (**B**) was analyzed using an AlphaScreen^®^SureFire^®^ kit (Perkin Elmer) and β-arrestin 2 recruitment (**C**) was determined by BRET. ERK1/2 phosphorylation and β-arrestin 2 recruitment data are expressed as increases in percentage over basal, whereas cAMP data are expressed as percentage with respect to values obtained with forskolin. Dynamic mass redistribution (DMR) tracings (**D**) represent the picometer-shifts of reflected light wavelength over time. All data are the mean ± S.E.M. of eight different experiments performed in triplicates. In cAMP (in the absence of forskolin), ERK1/2 phosphorylation, and β-arrestin 2 recruitment assays, one-way analysis of variance (ANOVA) followed by Bonferroni’s multiple comparison *post hoc* test were used for significance analysis. ** *p* < 0.01, *** *p* < 0.001, and **** *p* < 0.0001 *versus* vehicle treatment (basal); ^#^
*p* < 0.05 significance *versus* agonist treatment; ^&&&&^
*p* < 0.0001 *versus* forskolin treatment.

**Figure 4 ijms-21-05070-f004:**
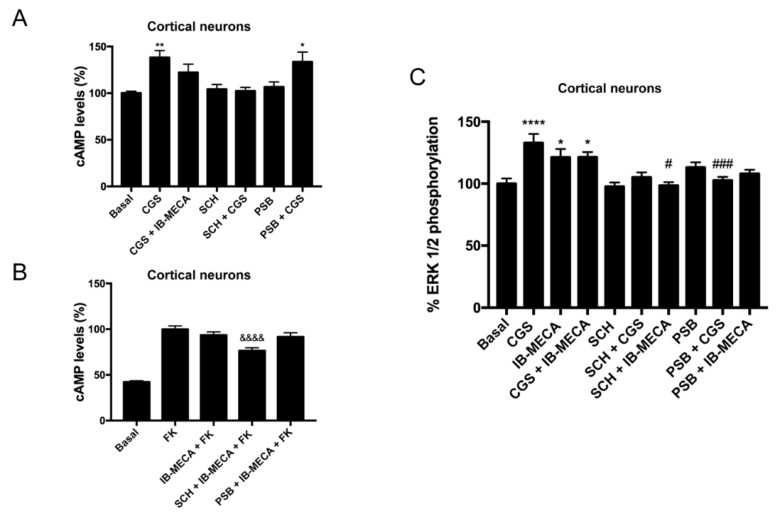
cAMP determination and ERK1/2 phosphorylation assays in primary cultures of cortical neurons. Panels (**A**,**B**): neurons cultured for 14 days were stimulated with selective receptor antagonists (1 µM SCH 442416 for A_2A_R or 1 µM PSB-10 for A_3_R) and subsequently with selective agonists (100 nM CGS 21680 for A_2A_R and 100 nM IB-MECA for A_3_R), then intracellular cAMP concentration was determined. In panel B, experiments were performed in cells finally treated with 500 nM forskolin. Values are the mean ± S.E.M. (n = 6 in triplicates). Panel (**C**): neurons cultured for 14 days were stimulated with selective antagonists for 10 min and subsequently with selective agonists for 7 min, then ERK1/2 phosphorylation was determined. Values are the mean ± S.E.M. (n = 6 in triplicates). One-way ANOVA followed by Bonferroni’s multiple comparison *post hoc* test were used for significance analysis. * *p* < 0.05, ** *p* < 0.01, and **** *p* < 0.0001 *versus* vehicle treatment (basal); ^#^
*p* < 0.05, ^###^
*p* < 0.001 *versus* agonist treatment. ^&&&&^
*p* < 0.0001 *versus* forskolin treatment.

**Figure 5 ijms-21-05070-f005:**
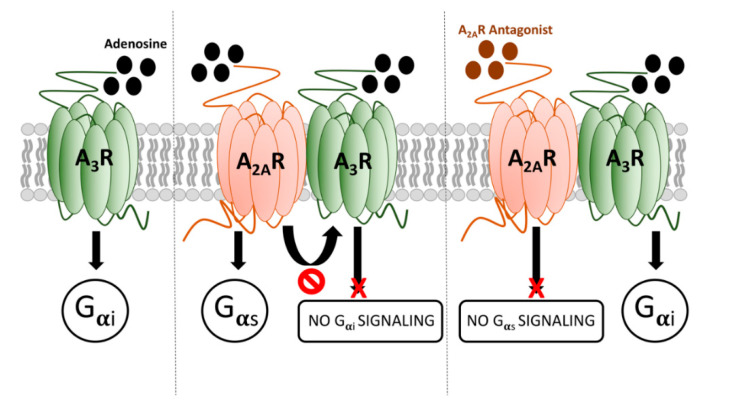
Scheme summarizing the signaling output of adenosine when interacting with A_2A_R/A_3_R-Hets.

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
