# Peer review of "Adenosine A2A and A3 Receptors Are Able to Interact with Each Other. A Further Piece in the Puzzle of Adenosine Receptor-Mediated Signaling"

_ijms, 2020, doi:10.3390/ijms21145070_

Round 1

Reviewer 1 Report

This manuscript found a novel interaction pattern of A2A and A3 receptor, which is important for investigating the function mechanism of adenosine receptors and also helpful drug design. The present results are interesting and the methods used are solid and supporting the finding. Only a few comments:

  1. it is better to add the units in Fig. 1.
  2. it will be great if the author could provide the reference about the " Adenosine receptors are divided into high affinity A1R and A2AR types, and low affinity A2BR and A3R types." in line 81.  

Author Response

Reviewer #1

This manuscript found a novel interaction pattern of A2A and A3 receptor, which is important for investigating the function mechanism of adenosine receptors and also helpful drug design. The present results are interesting and the methods used are solid and supporting the finding.

Response: we appreciate the positive comments.

Only a few comments:

  1. It is better to add the units in Fig. 1.

Response: Thanks for suggestion. This issue has been fixed in the revised version of the paper.

  1. It will be great if the author could provide the reference about the " Adenosine receptors are divided into high affinity A1R and A2AR types, and low affinity A2BR and A3R types." in line 81.

Response: we acknowledge this suggestion. As per request of the reviewer references to early reviews in the field have been added to the revised version of the manuscript.

Observation: Major changes are highlighted in yellow. Grammatical corrections are not highlighted.

Reviewer 2 Report

ARs are promising therapeutic targets in a wide range of conditions and as such are subject of study since many years. The difficulty when developing active ingredients is that each adenosine receptor affects not just one but a large number of body functions. Development of sufficiently selective AR modulators with the required pharmacological characteristics is an attractive but still unrealized goal. This is one of the reasons that after decades  of medicinal chemistry research and discovery of  number of more or less selective AR ligands none have yet become a widely used medication. These difficulties get more complicated by unwanted non-adenosine receptor drugs interactions and different ARs interactions, including discovered by authors A2A AR and A3 AR complex formation.

This is interesting work and valuable addition to ARs field, I recommend its publications after addressing the following concerns:

Abstract is inadequate for this article. It should contain brief information on why this research was undertaken, how it was done, what is the most important result and conclusion. In other words, abstract should relate to this work, and in present form it unfortunately resembles rather an abstract for a review.

Line 52: „No sign of adenosine receptor-mediated calcium signaling has been reported”

This is not correct, please check e.g.:

https://link.springer.com/article/10.1007/s00424-014-1529-8

https://onlinelibrary.wiley.com/doi/full/10.1111/apha.12473

Line 79 and following. Please explain or hypothesize why do both A2 and A3 receptors locate as a heterodimer in the membrane of transfected cells ? What is the driving force?

Line 97 and following. A2A AR agonist CGS 21680 (Ki = 27 nM) has affinity for both A1 and A3 adenosine receptors but can be used to distinguish A2A- and A2B-mediated effects. Therefore, a direct interaction of CGS 21680 with the A3 receptor (in the concentration range used by the authors) cannot be excluded. What exactly indicates the allosteric mechanism of A2R regulation in the A3 neighborhood? Please discuss.

Line 124. “Taking into account these facts, confirmed in cells only expressing one of the receptors, we measured cAMP levels in HEK-293T cells”. It should be “over-expressing”.

Figure 3. DMR abbreviation should be explained. Short introduction to DMR technique should be provided in line 163.

Figure 4. The lack of an effect of IB-MECA on forskolin- and CGS 21680-induced cAMP level in neurons is puzzling. What is the expression of A3R in these cells?

Figure 5. Unclear message. It is not known what the authors would like to communicate. Incorrect description.

Line 217. The sentence “A simple scheme summarizing the operation of A1-A2A, A2A-A2B or A2A-A3 receptor heteromers is provided in Figure 5” does not match the Figure 5.

The close proximity of receptors in the cell membrane can be demonstrated by structural studies. The easiest way is to study co-localization with the use of confocal or electron microscopy. Given the excellent fluorescence methodology available to the authors, such additional research could quickly verify the A2A-A3 heterodimer hypothesis and is strongly advised.

Line 231. TLB abbreviation should be explained.

What is the expression pattern of adenosine receptors on HEK-293T parental cell line? The sequences of cDNAs should be provided.

Did authors perform control transfection to check for cell health? Lack of information on it in the chapter 4.3.

Author Contributions.  There is no description of I.R.R.'s contribution.

Author Response

Reviewer #2

ARs are promising therapeutic targets in a wide range of conditions and as such are subject of study since many years. The difficulty when developing active ingredients is that each adenosine receptor affects not just one but a large number of body functions. Development of sufficiently selective AR modulators with the required pharmacological characteristics is an attractive but still unrealized goal. This is one of the reasons that after decades of medicinal chemistry research and discovery of number of more or less selective AR ligands none have yet become a widely used medication. These difficulties get more complicated by unwanted non-adenosine receptor drugs interactions and different ARs interactions, including discovered by authors A2A AR and A3 AR complex formation.

Response: we appreciate the clear and succinct summary of the paper.

This is interesting work and valuable addition to ARs field, I recommend its publications after addressing the following concerns.

 Response: we appreciate the positive comment.

Abstract is inadequate for this article. It should contain brief information on why this research was undertaken, how it was done, what is the most important result and conclusion. In other words, abstract should relate to this work, and in present form it unfortunately resembles rather an abstract for a review.

Response: thanks for the suggestion. As per reviewer’s request, we have prepared a new abstract for the revised version of the manuscript.

Line 52: „No sign of adenosine receptor-mediated calcium signaling has been reported”

This is not correct, please check e.g.:

https://link.springer.com/article/10.1007/s00424-014-1529-8

https://onlinelibrary.wiley.com/doi/full/10.1111/apha.12473

Response: thanks for raising this issue. It is true that adenosine receptors may mediate regulation of calcium signaling, but not via Gq coupling. We have rewritten sentences and included these two and a third reference in the revised version of the paper.

Line 79 and following. Please explain or hypothesize why do both A2 and A3 receptors locate as a heterodimer in the membrane of transfected cells ? What is the driving force?

 Response: we acknowledge this comment. We have provided information, in the revised version of the manuscript, about the potential driving force as per analogy with another heteromers (A1-A2A and the A2A-A2B) for whom a structural model has been recently proposed.

Line 97 and following. A2A AR agonist CGS 21680 (Ki = 27 nM) has affinity for both A1 and A3 adenosine receptors but can be used to distinguish A2A- and A2B-mediated effects. Therefore, a direct interaction of CGS 21680 with the A3 receptor (in the concentration range used by the authors) cannot be excluded. What exactly indicates the allosteric mechanism of A2R regulation in the A3 neighborhood? Please discuss.

 Response: We have addressed this issue in the revised version of the manuscript contemplating the possibility that CGS21680 may interact to the A3 receptor. A key factor is that A2A activation engages Gs whereas A3 activation engages Gi.

Line 124. “Taking into account these facts, confirmed in cells only expressing one of the receptors, we measured cAMP levels in HEK-293T cells”. It should be “over-expressing”.

Response: thanks for the comment but we do not like the term “overexpressing”. It is true that many experiments performed along years with GPCR expressed in heterologous system were overexpressed, in the “heteromer” field we try to reduce the expression to the levels that can be found in natural sources (for instance in the case of the A2A, the expression in transfected cells is lower than in cells in the striatum). In this sense, we have omitted the word “only” and we will refer to cells that are transfected with the plasmid for one of the two receptors.

Figure 3. DMR abbreviation should be explained. Short introduction to DMR technique should be provided in line 163.

 Response: we appreciate the suggestion that has been taken into account in the revised version of the paper. An ad hoc reference from a lab pioneering the use of this technique for GPCR research has been included in the revised version of the paper.

Figure 4. The lack of an effect of IB-MECA on forskolin- and CGS 21680-induced cAMP level in neurons is puzzling. What is the expression of A3R in these cells?

Response: thanks for the comment. Our interpretation based on similar results in transfected cells and in cortical neurons is that agonists are acting mainly via the A2A-A3 heteromer; the A3 is not functionally coupled to Gi when forming a complex with the A2AR.

Figure 5. Unclear message. It is not known what the authors would like to communicate. Incorrect description.

Line 217. The sentence “A simple scheme summarizing the operation of A1-A2A, A2A-A2B or A2A-A3 receptor heteromers is provided in Figure 5” does not match the Figure 5.

Response: thanks for detecting this mistake. We are preparing a commissioned review on adenosine receptor heteromers and we just cropped the whole figure to show only the part of the A2A-A3 heteromer. We have corrected the figure legend.

The close proximity of receptors in the cell membrane can be demonstrated by structural studies. The easiest way is to study co-localization with the use of confocal or electron microscopy. Given the excellent fluorescence methodology available to the authors, such additional research could quickly verify the A2A-A3 heterodimer hypothesis and is strongly advised.

 Response: this appropriate suggestion has been considered; colocalization images have been included in the revised version of the manuscript.

Line 231. TLB abbreviation should be explained.

 Response: we appreciate detecting this missing item. As per request of the reviewer, we have addressed this issue and a description of this abbreviation is now included in the revised version of the manuscript.

What is the expression pattern of adenosine receptors on HEK-293T parental cell line? The sequences of cDNAs should be provided.

Response: we have referred to the sequences deposited in the databases (using the ref number to avoid misinterpretations due to polymorphisms), and the cloning site (in the case of fusion proteins) was specified in the paper. Sequencing is always performing with all constructs used in a given research project. This extra information has been included in the revised version of the paper.

Did authors perform control transfection to check for cell health? Lack of information on it in the chapter 4.3.

 Response: we acknowledge this suggestion. The requested information has been added in the revised version of the paper.

Author Contributions.  There is no description of I.R.R.'s contribution.

Response: thank you very much for detecting this forgetfulness (fixed in the revised version of the paper).

Observation: Major changes are highlighted in yellow. Grammatical corrections are not highlighted.